# Attenuating Colorectal Cancer Using Nine Cultivars of Australian Lupin Seeds: Apoptosis Induction Triggered by Mitochondrial Reactive Oxygen Species Generation and Caspases-3/7 Activation

**DOI:** 10.3390/cells12212557

**Published:** 2023-10-31

**Authors:** Kishor Mazumder, Asma Aktar, Sujatha Ramasamy, Biswajit Biswas, Philip G. Kerr, Christopher Blanchard

**Affiliations:** 1Department of Pharmacy, Jashore University of Science and Technology, Jashore 7408, Bangladesh; 2School of Optometry and Vision Science, UNSW Medicine, University of New South Wales (UNSW), Sydney, NSW 2052, Australia; 3Department of Pharmacy, Dhaka International University, Dhaka 1212, Bangladesh; 4Institute of Biological Sciences, Faculty of Science, Universiti Malaya, Kuala Lumpur 50603, Malaysia; 5Institute for Molecular Bioscience, Queensland University, Brisbane, QLD 4072, Australia; 6School of Biomedical Sciences and Graham Centre for Agricultural Innovation, Charles Sturt University, Boorooma St., Wagga Wagga, NSW 2650, Australia

**Keywords:** colon cancer, lupin, apoptosis, polyphenol, caspases-3/7, intracellular ROS

## Abstract

As Australian lupin cultivars are rich sources of polyphenols, dietary fibers, high-quality proteins, and abundant bioactive compounds with significant antioxidant, antidiabetic, and anticancer activities, this research work is aimed at investigating the colon cancer alleviation activity of nine cultivars of lupin seeds on HCT116 and HT29 colon carcinoma cell lines through anti-proliferation assay, measurement of apoptosis, and identification of the mechanism of apoptosis. Nine cultivars were pre-screened for anti-proliferation of HCT116 and HT29 cells along with consideration of the impact of heat processing on cancer cell viability. Mandelup and Jurien showed significant inhibition of HCT116 cells, whereas the highest inhibition of HT29 cell proliferation was attained by Jurien and Mandelup. Processing decreased the anti-proliferation activity drastically. Lupin cultivars Mandelup, Barlock, and Jurien (dose: 300 μg/mL) induced early and late apoptosis of colon cancer cells in Annexin V-FITC assay. The mechanism of apoptosis was explored, which involves boosting of caspases-3/7 activation and intracellular reactive oxygen species (ROS) generation in HCT116 cells (Mandelup and Barlock) and HT29 cells (Jurien and Mandelup). Thus, the findings showed that lupin cultivars arrest cell cycles by inducing apoptosis of colorectal carcinoma cells triggered by elevated ROS generation and caspases-3/7 activation.

## 1. Introduction

Lupin, considered a super food for humans, is a promising member of the *Fabaceae* family. It belongs to leguminous pulses as a potential human food ingredient due to having a high content of antioxidants, essential amino acids, minerals, proteins (~40%), and dietary fibres (~28%) [1]. Because of being a leguminous pulse, lupin plays important role in the management of cropping system through crop rotation as well as nitrogen fixation. Although 450 species of lupin are grown worldwide, about 164 species have been cultivated agro-economically as food substances, as documented in the Integrated Taxonomic Information System [2]. In Australia, the Mediterranean climate and acidic nature of sandy soils are favorable to cultivate sweet lupin, among which narrow-leafed species, *Lupinus angustifolius* Blanco, white species, and *Lupinus albus* L. are economically and nutritionally considerable as human foods [3,4]. Lupin-based studies have been expanded progressively due to its significant goodness for human health. It exhibits multiple pharmacological activities, including antioxidant, antidiabetic, anti-hypertensive, cardioprotective, antihyperlipidemic, renoprotective, anticancer, anti-tumor, gastroprotective, and antimicrobial effects, which have been reported by researchers in varieties of in vivo studies [5,6,7,8].

Currently, cancer is considered a global burden for being the second prominent cause of mortality. As of 2019, risk-attributable deaths caused by different types of cancers had been increased by 20.0% [9]. The third most frequent disease and the fourth most common cause of cancer-related death is colorectal cancer, which is majorly affected by numerous factors, such as patients’ age, history of chronic diseases, and lifestyle [10,11,12]. A critical predictive factor for colorectal cancer survival in both men and women is the clinical stage when the disease is diagnosed. Case studies based on the clinical stage during diagnosis demonstrated that the 5-year relative survival rates for males and females with colorectal cancer, respectively, vary from 11.9% to 12.9% in stage IV and from 91.8% to 91.3% in stage I. The estimated 5-year relative survival rates based on the pathological stage are 91.2% and 96.2% in stage I and 19.1% and 19.8% in stage IV, respectively [13]. Based on clinical stages and the nature of colorectal cancer, a number of treatment strategies have been developed, such as a multimodal approach comprising surgical resection followed by chemotherapy with monoclonal antibodies or proteins against vascular endothelial growth factor and epidermal growth receptor [14,15]. Furthermore, alternative therapies, including the use of anti-inflammatory medications, probiotics, and gold nanoparticle-based pharmaceuticals, are now under research to improve treatment effectiveness and lessen side effects compared to conventional chemotherapy [16,17].

The choice of suitable treatment option depends solely on the disease states of colorectal cancer (resectable, borderline, locally advanced, and metastatic) and demands combination therapy including surgery, chemotherapy, chemo-radiotherapy, and supportive care [18]. Chemotherapy has various shortcomings, including adverse effects on healthy cells, high relapse and recurrences, and difficulties in treating and managing cancer due to chemo-resistance [19]. Plant-derived bioactive compounds possess multiple bio-functionalities by functioning as antioxidants, immunomodulators, antitumor actors, and chemoprotectors. Moreover, lupin seeds are evidenced to contain significant amounts of dietary fibers that act as prebiotics, which have additional beneficial effects on intensifying gut microflora to strengthen microenvironment homeostasis, condense toxic metabolites, and alter harmful immune reactions in the intestine, leading to the modification of epigenetic factors of colorectal cancer [1,20,21].

Considering these aspects, as the anticancer activity of lupin cultivars against colorectal cancer has not yet been reported, this research aimed to investigate the colorectal cancer alleviation activity of seed flours from nine cultivars of Australian lupin along with elucidating their mechanistic insights and evaluating the impact of heat processing on their activity.

## 2. Materials and Methods

### 2.1. Chemicals

Methanol, isopropanol, n-hexane, and dimethyl sulfoxide (DMSO) were purchased from Merck (Darmstadt, Germany). Dulbecco’s modified eagle’s medium (DMEM), fetal bovine serum, penicillin, streptomycin, sodium pyruvate, Annexin V FITC, propidium iodide, dichloro-dihydro-fluorescein di-acetate (DCFH-DA), tris-hydrochloride buffer, EDTA, and trypsin solution were obtained from Sigma-Aldrich (St Luis, MO, USA). All of the reagents, solvents, and standards used in this study were of high-purity analytical grade.

### 2.2. Raw Materials

Whole seeds of Australian lupin cultivars were received as gift samples from the NSW Department of Primary Industries, Wagga, NSW, Australia. Three cultivars from *L. albus* species (WK-338, Luxor, and Rosetta) and six cultivars from *L. angustifolius* species (Jenabillup, Mandelup, Barlock, Jindulee, Gunyidi, and Jurien) were investigated in this study.

### 2.3. Processing of Lupin Seeds

Whole seeds of Australian lupin cultivars were dehulled by a dehuller (Abrasion Debranner VTA5, Satake, Australia) and separated to seed coats and kernels through vacuum separator. The separated kernels were dried (45–50 °C) and grinded using a grinder (Magic Bullet, Santos 01PV, Nella, France). Finally, the powdered seed flours were sieved, vacuum packed, and stored at 4 °C for further use [2,7].

### 2.4. Extraction of Lupin Seed Flour

Lupin seed flour was extracted through the method of Mazumder et al. [2,7]. Briefly, kernel flours from each cultivar were divided equally in two portions. One portion was taken with three-fold water followed by boiling (100 °C) for 30 min. The processed flours were then dried (60 °C, 3 h) and extracted. Both of the processed and unprocessed flours of kernels were defatted by mixing with ten-fold solvent (hexane and isopropanol, 3:2 *v*/*v*) followed by overnight stirring, centrifugation (1792 g, 10 min), and drying at room temperature. After defatting, kernel flours were extracted with 80% methanol through the same sequential process of extraction used for the extraction of seed coats of lupin cultivars.

### 2.5. Assessment of Attenuating Effects on Colon Cancer

To analyze the anticancer activity of Australian lupin seed flours, two human colorectal cancer cell lines, HCT116 (wild-type p53) and HT29 (mutant p53), were selected based on their differential morphology as well as their malignant characteristics, simulating primary-staged and advanced-staged colon carcinoma cells, respectively [22]. To attain preliminary approximation of anticancer activity, nine cultivars of lupin seed flour extracts (both processed and unprocessed) were assayed for inhibition of proliferation and cell viability by MTT assay. The cultivars showing higher inhibition of proliferation were selected for further evaluation of cell cycles as well as quantification of apoptosis. Finally, the cultivars with the best apoptotic activities were tested for exploration of mechanistic insights by analyzing the intracellular reactive oxygen species (ROS) and caspase-3/7 activation activity. Figure 1 represents the summary of the overall workflow of the research work.

#### 2.5.1. Cell Culture

Two different colon cancer cell lines, HCT116 and HT29, were cultured differently in Dulbecco’s modified eagle’s medium (DMEM), which was supplemented with fetal bovine serum (10%), penicillin–streptomycin mixture (final concentration: 100 U/mL and 100 µg/mL, respectively), and sodium pyruvate (1%). A humid atmosphere (95% air + 5% CO_2_) was maintained at 37 °C to ensure optimal growth of the cells [22].

#### 2.5.2. Cell Viability Assay

The cytotoxicity of lupin seed flour extracts on human colorectal HCT116 and HT29 cells was assayed according to the MTT method reported by Batoul Al-Khatib et al. [23]. This assay is based on the formation of crystalline formazan as a result of cancer cell metabolism of tetrazolium salt (3-(4,5-dimethylthiazol-2-yl)-2,5-diphenyl tetrazolium bromide). As per the method, the cancer cell suspensions (density: 1 × 10^4^ cells/mL) were seeded in 96-well plates followed by adjusting the volume of each well to 200 µL with medium. The wells containing medium and cells were allowed to adhere by pre-incubation for 24 h. Then, the wells were washed with phosphate-buffered saline and treated with lupin flour extracts with varying concentrations of 1.0–1000 μg/mL followed by 48 h of incubation at 37 °C. Additional wells were prepared as controls by adding DMSO to the washed well instead of extracts. After the post-treatment period, the wells were further incubated for 3 h in 100 μL/well MTT solution (Sigma-Aldrech, Darmstadt, Germany). After the indicated incubation period, synthesized formazan product was stabilized using a solubilizing solution (prepared through the addition of 90 μL of 100% DMSO and 60 μL of 30% SDS solution) in each well and incubated in the dark for 1 h [24]. Finally, the optical density (OD) was read at 570 nm (with a reference of 690 nm) with an Enzyme-Linked Immunosorbent Assay (ELISA) reader (ThermoScientific, Waltham, MA, USA) where the solubilizing solution was considered blank. Data were expressed as percent viability of cell along with IC_50_ values. The viability percentage was calculated according to the following Equation (1).
(1)$$\%\;Viability=\frac{{\left(Mean\;OD\right)}_{Sample}-{\left(Mean\;OD\right)}_{Blank}}{{\left(Mean\;OD\right)}_{Control}-{\left(Mean\;OD\right)}_{Blank}}\times 100$$

#### 2.5.3. Quantification of Apoptosis through Annexin V Assay

For the measurement of apoptosis, four cultivars (Barlock, Jenabillup, Jurien, and Mandelup) against HCT116 colorectal carcinoma cells and two cultivars (Jurien, and Mandelup) against HT29 colon carcinoma cells were selected based on their higher inhibition of cell proliferation in the cell viability assay. Due to the drastic loss of anti-proliferating activity of all of the cultivars of lupin except WK-338, they were excluded from further studies. As, overall, the anti-proliferating activity of WK-338 seeds (both of the processed and unprocessed flours) was below 50%, they were also omitted for further analysis. HCT116 and HT29 cells were treated with unprocessed lupin seed flour extracts (dose: 300 μg/mL) for 24 h. After 24 h, the media from each well were stored in tubes and the wells with cancer cells were washed (once with PBS followed by twice with trypsin) and incubated (for 5 min in 37 °C). In 1 mL of PBS, the trypsinized cancer cells were collected and transferred to distinct tubes containing media for each well. The cells were then resuspended in binding buffer (cell density: 1 × 10^4^ cells/mL), and Annexin V FITC (1 µL per 100 µL cell suspension) and propidium iodide (2 µL per 100 µL cell suspension) were added followed by incubation for 15 min. Finally, the cell suspensions were analyzed in the flow cytometer (BD Biosciences, Heidelberg, Germany). Triplicate readings for each well were measured [25].

#### 2.5.4. Reactive Oxygen Species Detection

Mitochondrial ROS generation is considered one of the provoking factors for initiating apoptosis in cancer cells. HCT116 and HT29 cells were seeded in a 24-well plate (cell density: 1.5 × 10^4^ cells per well) and incubated for 24 h. The cells were washed and treated for a further 24 h with lupin seed extracts. To analyze mitochondrial ROS, 10 μmol/L ROS-sensitive dye dichloro-dihydro-fluorescein di-acetate (DCFH-DA) was added to the treated HCT116 and HT29 cells and incubated for 30 min at 37 °C. During the incubation period, the cells incorporated DCFH-DA and oxidized to 2,7-dichlorofluorescein (DCF) through mitochondrial ROS. The free dye was removed by washing with ice-cold PBS. Finally, the fluorescence intensities of DCF were read through flow cytometry (488 nm laser and detector 528 nm) [26].

#### 2.5.5. Caspase-3/7 Detection Assays

HCT116 and HT29 cells were treated with lupin seed extracts for 24 h. After the treatment period, the cells were lysed by treating with tris-hydrochloride buffer pH 8.0 (0.05 M), EDTA (0.01 M), SDS (1%), and enzyme inhibitors (protease inhibitors and phosphatase inhibitors) followed by sonication and subsequent centrifugation (10 min, 4032 g). The supernatant was analyzed for concentrations of protein by utilizing a BCA protein assay kit (Beyotime, Shanghai, China). Protein samples were subjected to electrophoresis (6% SDS-PAGE gel fast preparation kit) (Epizyme, Shanghai, China) followed by subsequent transfer to pre-incubated polyvinylidene-difluoride (PVDF) membranes with antibodies against caspase-3, caspase-7, cleaved caspase-3, and cleaved caspase-7 proteins [27].

### 2.6. Statistical Analysis

Data were analyzed through one-way ANOVA with subsequent Dunnett’s post hoc statistical tests for multiple comparisons performed in GraphPad Prism 6.0 (GraphPad Software, La Jolla, CA, USA). Student’s *t*-test was utilized to compare between two groups as well as to examine statistical significance. Data were expressed as mean ± standard error of mean (SEM), and *p* < 0.05 was considered statistically significant.

## 3. Results

### 3.1. Concentration-Dependent Cytotoxic Effects of Lupin Seed Flours on Colon Cancer

Unprocessed lupin seed flour extracts showed concentration-dependent inhibition of both the HCT116 and HT29 colorectal cancer cells’ proliferation. The findings of the cytotoxicity of lupin seed flour extracts on HCT116 cells were presented in Figure 2. Among the nine cultivars, Barlock, Gunyidi, Jenabillup, Jindulee, Jurien, Mandelup, and WK-338 inhibited HCT116 cells’ proliferation. Unprocessed flour extract of Mandelup showed the highest inhibition (87.76 ± 4.33%, IC_50_: 262.7 μg/mL) of HCT116 cells, whereas WK showed the least (33.40 ± 2.99%) inhibition. Unprocessed flour extracts of Barlock, Jenabillup, Jurien, and Mandelup inhibited >60% of HCT116 colorectal cancer cells with IC_50_ values ranging around 180~300 μg/mL. Among these cultivars, Jurien showed inhibition with the least IC_50_ value (181.9 μg/mL). In addition, about >80% of cell death was attained by unprocessed flour extracts of Barlock and Mandelup cultivars. Inter-species comparison between the cultivars of the two species of lupin *L. angustifolius* and *L. albus* demonstrated that six studied cultivars of *L. angustifolius* showed anti-proliferative activity against colorectal cancer cells, whereas cultivars of *L. albus* Luxor and Rosetta failed to show cytotoxicity to HCT116 cells. WK-338, a cultivar of *L. albus*, was shown to exhibit mild cytotoxicity (<40% inhibitions of cell proliferation) to colorectal cancer cells. These findings demonstrate that lupin seed flour extracts, especially four cultivars from *L. angustifolius* (Barlock, Jenabillup, Jurien, and Mandelup) species of Australian lupin, have potential cytotoxicity on primary-staged colorectal cancer cells, like HCT116.

In the case of HT29 colorectal cancer cells, amongst the nine cultivars, five of them (Barlock, Gunyidi, Jenabillup, Jurien, and Mandelup) showed inhibition of proliferation ranging around 42–79%. The highest inhibition of proliferation was attained by Jurien (79.34 ± 5.31%, IC_50_ value: 285.7 μg/mL), whereas Barlock showed the least inhibition (42.32 ± 4.11%). Greater than 60% inhibition of HT29 cell proliferation was obtained by Jurien and Mandelup (IC_50_ values of 221.4 and 285.7 μg/mL, respectively). Jindulee, a cultivar of *L. angustifolius*, and three studied cultivars of *L. albus* (WK-338, Luxor, and Rosetta) did not show any cytotoxicity to HT29 colorectal carcinoma cells at the concentration of 1000 μg/mL. The obtained results are shown in Figure 3, which specifies that Australian lupin cultivars Jurien and Mandelup have significant cytotoxicity on advance-staged colorectal cancer cells, like HT29.

### 3.2. Impact of Processing of Lupin Flours on Inhibition of Colon Cancer Cells’ Proliferation

Figure 2E and Figure 3C–E represent the impact of domestic cooking of lupin seed flour extracts on the inhibition of colorectal cancer cells’ proliferation. The unprocessed and processed flour extracts were assayed to quantify colorectal cancer cells’ proliferation, and the obtained cytotoxic activities were compared using Student’s *t*-test. Regarding HCT116 colorectal cancer cells, significant loss of cytotoxic activity was observed for all of the nine cultivars except WK-338 (Figure 3D). After processing, Barlock and Mandelup were shown to harshly lose (69.13 ± 4.66% and 79.12 ± 2.63%, respectively) cytotoxic activities, which were statistically significant (*** *p* < 0.001) compared to the activities of unprocessed flours of each cultivar. The significant reduction in anti-proliferation activity was also observed in the case of Jindulee (** *p* < 0.01) and Jurien (* *p* < 0.05). Conversely, WK increased 10.80 ± 0.88% of the inhibition of HCT116 cell proliferation after processing. Luxor and Rosetta, cultivars of *L. albus*, were also failed to show cytotoxicity to the studied cell lines in processed conditions. Though the activity was reduced drastically, all of the cultivars of *L. angustifolius* showed mild cytotoxicity to HCT116, like primary-stage colorectal cancer cells.

Interestingly, similar commonness of reduction in cytotoxic activity of lupin flour was observed on HT29 cells after the processing of seed flours (Figure 3E). All five of the cultivars that showed inhibition of HT29 cell proliferation in an unprocessed state harshly lost their cytotoxic activities by around 16~58%. Mandelup lost the maximum cytotoxic activity (58.10 ± 4.59%), which is statistically significant (*** *p* < 0.001) compared to the cytotoxicity of the unprocessed flour extract of that cultivar. Moreover, Barlock, Gunyidi, and Jurien diminished the potentiality of inhibiting HT29 cell proliferation (* *p* < 0.05) as well. Likewise, unprocessed flour extracts of a cultivar of *L. angustifolius*, Jindulee, and all three of the cultivars of *L. albus* failed to induce cytotoxicity in HT29 colorectal cancer cells in the preceding condition.

The loss of cytotoxicity among the cultivars did not follow any considerable patterns. Loss of activity of Mandelup was unexpectedly very extreme (>60%) for both of the cell lines compared to activities of other cultivars, like Gunyidi (<20%), Jenabillup (<15%), and Jurien (~20%), which might be considered slight to moderate. Barlock, another cultivar of *L. angustifolius*, was shown to lose activity tremendously (~65%) to HCT116 cells but marginally (~20%) to HT29 cells. Therefore, the loss of activity of the processed flours of lupin was not correlated to any inter-species- or inter-cultivar-related factors. Thus, from the findings, it is revealed that lupin seed flours lost their cytotoxic activity enormously to both of the primary- and advanced-stage colorectal cancers cells due to heat processing as well as domestic cooking.

### 3.3. Induction of Apoptosis and Cell Cycle Arrest by Lupin Seed Flour Extracts

Lupin cultivars induced apoptosis in both of the HCT116 and HT29 cells. Annexin V-FITC assay results (Figure 4) showed that four cultivars (Mandelup, Barlock, Jurien, and Jenabillup (selected from MTT cytotoxicity assay)) induced apoptosis of HCT116 cells after 24 h of treatment with 300 μg/mL of unprocessed extracts. Mandelup and Barlock significantly induced the accumulation of HCT116 colorectal cancer cells in the apoptotic (early apoptosis and late apoptosis) phase. Among these four cultivars, Mandelup induced the highest induction of apoptosis (total: 35.80 ± 2.1%; early: 7.6 ± 0.9%; and late: 28.2 ± 1.2%) of HCT116 cells, which was statistically significant compared to the other cultivars. Alternatively, Jenabillup showed the least apoptotic activity (total: 9.40 ± 2.52%) to HCT116 cells. As regards HT29 colorectal cancer cells, data obtained from flow cytometry (Figure 4I) demonstrated that Jurien and Mandelup increased the induction of apoptosis of advanced colorectal cancer HT29 cells. Jurien showed the higher induction of apoptosis (total: 34.30 ± 4.2%; early: 16.80 ± 1.8%; and late: 17.50 ± 2.4%), which is statistically significant (* *p* < 0.05) compared to that of Mandelup. From these findings, it is found that lupin seed flour (unprocessed) extracts increased the induction of apoptosis of primary-stage as well as advanced-stage colorectal cancer cells.

### 3.4. Generation of Mitochondrial ROS by Lupin Seed Flour Extracts

As the formation of mitochondrial ROS is considered one of the potential triggers to induce apoptosis in cancer cells, ROS formation in HCT116 and HT29 cells due to treatment with lupin seed flour extracts has been investigated in this study. After 24 h of treatment with Mandelup (300 μg/mL), significant (* *p* < 0.05) elevation in the formation of ROS was observed in HCT116 colorectal cancer cells compared to untreated cells. Apart from the 3.23-fold increase in the oxidized DCF level, fluorescent images confirmed the mitochondrial ROS generation after treatment with unprocessed flour extract of Mandelup. Furthermore, Barlock showed a 1.70-fold increase in the oxidized DCF level compared to untreated carcinoma cells. Concerning HT29 cells, similar findings were noticed after 24 h of treatment with Jurien and Mandelup seed flour extracts. Approximately 2.11-fold and 1.30-fold upgrades in the oxidized DCF levels in Jurien and Mandelup, respectively, and fluorescent images demonstrated that Jurien as well as Mandelup seed flour extracts elevated ROS generation in the mitochondria of HT29 colorectal carcinoma cells. These results are presented in Figure 5, which establishes that seed flours of Mandelup, Jurien, and Barlock cultivars of Australian lupin upgrade ROS formation in the mitochondria of carcinoma cells, leading to the triggering of the induction of apoptosis of the primary-stage as well as the advanced-stage colorectal carcinoma cells.

### 3.5. Activation of Caspase-3/7 by Lupin Seed Flour Extracts

The apoptosis of cancer cells is mediated by the activation of caspases-3/7 enzymes as a central cofactor. Inactive procaspases require cleaving to be functional as active caspases. Therefore, in this study, the activation of caspases-3/7 enzymes in colorectal cancer cells by lupin seed flour extracts was assayed through Western blotting to identify the underlying mechanism of apoptotic activity, and the results are presented in Figure 6 (Appendix A). After 24 h of treatment of unprocessed flours of Mandelup and Barlock (dose: 300 μg/mL), up-regulation of caspase-3 and caspase-7 cleavages was observed in HCT116 colorectal carcinoma cells. Mandelup significantly increased by 2.4-fold (* *p* < 0.05) and 2.1-fold (* *p* < 0.05) the cleavages of caspase-3 and caspase-7, respectively, compared to the untreated HCT116 colorectal cancer cells. Similarly, Barlock upgraded by 1.4-fold and 1.6-fold the cleavages of caspase-3 and caspase-7, respectively, which were considerably higher than those of the control (untreated cells) but not statistically significant. Treatment of HT29 colorectal cancer cells with Jurien seed flour extracts resulted in a significant increase in caspase-3 (2.1-fold) and caspase-7 (1.9-fold) cleaving in HT29 cancer cells, as well. Moreover, Mandelup also augmented the cleavage of the caspase-3 and caspase-7 enzymes (1.6- and 1.7-fold, respectively) in HT29 cells. Thus, the findings proved that lupin cultivars, especially Mandelup, and Jurien seed flour extracts exhibited apoptotic activity by increasing caspases-3/7 cleavage in colorectal carcinoma cells.

## 4. Discussion

Australian lupin species *L. angustifolius* (narrow-leafed lupin) and *L. albus* (white lupin) are leguminous seeds rich in polyphenols, flavonoids, tannins, essential amino acids, proteins, healthy fats, minerals, and dietary fibres [5]. Due to their abundant bioactive metabolites, lupin seeds possess numerous bio-functionalities; as such, they are considered a super food [7,28]. Various cultivars from *L. angustifolius* and *L. albus* species have been reported for attenuating activity against colorectal cancers [29,30,31], breast cancer [32,33], and hepatic carcinoma [34]. Australian lupin seeds are rich in polyphenols, alkaloid, and insoluble dietary fibers, which may produce a unique combination with the synergistic bioactivity required for side-effect-free management of colorectal cancers. Apart from these characteristics, Australian lupin cultivars from *L. angustifolius* and *L. albus* species are rich in antioxidants [2], which facilitate the balance of hemostasis in the cellular microenvironment and prevent cancers. Comprehensive analysis of phytocompounds present in lupin cultivars demonstrated that lupanine, hexadecanoic acid methyl ester, methyl stearate, docosenamide, octadecenoic acid methyl ester, and octadecadienoic acid methyl ester are produced in all of the nine cultivars of lupin. Quinolizidine alkaloids, like lupanine, 13-OH-lupanine, and 13α-acetoxy lupanine, are reported as one of the major phytochemicals found specifically in lupin cultivars exhibiting potential anticancer activity [35]. Lupin cultivars are also rich in esters of fatty acid, which are reported as potential defenders against cancer [36]. In addition, amino acids, anethole, campesterol, and chromo-peptides, like actinomycin C2, are also identified in lupin cultivars [2]. Actinomycin C2, a major antimicrobial component, has been reported for anti-tumor activity [37]. Phenolic compounds, a large group of a prime antioxidant family present in Australian lupin seeds, are phenolic acids, like caffeic, p-coumaric, ferulic, rosmarincic, chlorogenic, vanillic, protocatechuic, and p-hydroxybenzoic acid. Moreover, lupin cultivars are rich sources of flavonoid, flavonol, and flavones, such as quercetin, rutin, kaempferol, luteolin, apigenin, diosmetin, myricetin, and isoflavones [6,38,39]. Surprisingly, natural polyphenols possess the unique feature of performing dual functionalities through the neutralization of ROS at the period of cancer prevention in healthy cells and the generation of excess ROS in developed cancer cells to trigger apoptosis and cancer cell death [40]. Apart from these characteristics, protein isolates of lupin cultivars reduced metastasis of cancer cells by inhibiting the activity of MMP-9 enzyme [41]. Ahmed et al. narrated a surprising observation that exposed that lupin seed extract showed better apoptotic activity against colorectal cancer CACO-2 cell lines than an existing market drug, 5-fluorouracil [42]. Moreover, natural dietary fibers (also known as prebiotics) have significant roles in alleviating colorectal carcinoma by balancing microbiota present in the large intestine. The restoration of gut microflora helps to balance microenvironment homeostasis, neutralize toxic metabolites produced by cancer cells, and defend against undesirable immune reactions in the large intestine resulting in the modification of epigenetic factors of colorectal cancer [21,43].

Hence, in this investigation, nine cultivars of Australian lupin (Jenabillup, Mandelup, Barlock, Jindulee, Gunyidi, Jurien, WK-338, Luxor, and Rosetta) were analyzed to identify their attenuating activity on the primary-stage colorectal cancer cell line HCT116 and the advanced-stage colorectal carcinoma cell line HT29.

Lupin cultivars showed significant potential to reduce cell viability and to increase the inhibition of the proliferation of HCT116 and HT29 colorectal cancer cells. Among the nine cultivars, seven of them showed cytotoxicity to HCT116 cells, while five of them showing cytotoxicity to HT29 cells. Furthermore, the IC_50_ values for the inhibition of cell proliferation were found to be <300 μg/mL, which supported the potential role of lupin seed flours to combat colorectal cancer. As reported earlier, GC-MS phytochemical screening of lupin cultivars showed that lupin cultivars contained alkaloids, tannins, saponins, and esters of fatty acids [2]. Plant-derived alkaloids, tannin, tannic acids, and saponins have been reported to reduce cancer cell viability [44,45]. The pathogenesis of colorectal cancer has revealed that multiple molecular abnormalities are responsible for the occurrence of carcinogenicity and the progression to an advanced stage. Thus, multiple molecular targets are considered, and combination therapies have been adopted to combat colorectal carcinoma proliferation [15,46]. Surprisingly, due to the presence of multiple bioactive secondary compounds in lupin cultivars, synergistic effects might be exerted on reducing cancer cell proliferation, thus contributing to the observation of significant inhibition of HCT116 and HT29 colorectal cancer cells’ proliferation. It has also been reported that lupin seeds are surprisingly non-toxic to healthy cell lines [34]. Thus, lupin seeds showed selective cytotoxicity to cancerous cells.

Domestic heat processing of lupin seed flours reduced their anti-proliferative activities drastically. Among the nine cultivars, the cytotoxicity of all of the cultivars was reduced by many folds to both of the cell lines, with the exception of WK-338 to HCT116 cells. Interestingly, an increase of approximately 10% of the cytotoxic activity of WK-338 might be due to numerous possible reasons, such as (i) the presence of bioactive heat stable metabolites with anticancer activities, (ii) the presence of multiple bioactive compounds capable of undergoing heat-mediated chemical reactions resulting in synergism, and (iii) heat-triggered chemical reactions among metabolites, resulting in products with enhanced anticancer activity. There could be other possibilities that are not understood.

A tremendous reduction in the total flavonoids and flavonols in lupin cultivars was found after heat processing in a previous study [2]. Degradation of polyphenols by heat treatment has also been documented in many studies [47,48,49]. The functionality and potentiality of polyphenols against cancer cells depend on the structural stability of functional moiety of the responsible compounds. Heat treatment may degrade the functional groups of phenolic compounds responsible for anticancer activity. Apart from that, synergistic reactions responsible for the anticancer mechanism of polyphenols might be diminished many folds by heat [50]. Another possible reason behind the declining anticancer activity of lupin following heat treatment could be due to the loss of bioactive volatile compounds with anticancer activity during heating [51]. Sometimes, heat treatment can lead to undesirable chemical reactions among bioactive compounds, resulting in the formation of altered products lacking and/or antagonizing anticancer activity [2]. As the cytotoxic activity of all of the cultivars, excluding WK-338, was diminished significantly, the processed flours of lupin cultivars have been omitted for further investigation. WK-338 has also been excluded due to the moderate activity (<50%) of the processed and unprocessed flour extracts.

In general, the reduction of cancer cell viability is attributable to the results of apoptosis and/or necrosis. Apoptosis, a major type of programmed cell death, to some extent invariably determines the fate of malignant colorectal cancer cells in terms of development and progression [27]. To identify the underlying reasons behind the anti-proliferative activity of lupin cultivars, the cultivars with significant anti-proliferation activity were tested by analyzing the cell cycles and through the quantification of apoptosis. Among the nine cultivars, the unprocessed flour extracts of Mandelup and Barlock increased the induction of apoptosis (2–3 folds) of HCT116 cells significantly, whereas Jurien and Mandelup induced the apoptosis of HT29 cells. The cultivars affected both of the early and late apoptotic stages. The lupin cultivars Mandelup and Jurien are rich in phenolic acids, flavonoid, and flavonols. These natural compounds have the potential to induce apoptosis through differential mechanisms involving the activation of caspases cascade, the modification of oxidative-stress-induced mitochondrial toxicity, the activation of p53, and other mechanisms [52,53]. Furthermore, phytochemical profiles of lupin demonstrated that both of the cultivars, Mandelup and Jurien, contained lupanine, an alkaloid with significant anticancer activity that has been reported for triggering apoptosis by restoring p53 activation [2,15,54]. Moreover, proteins isolated from lupin have p53 restoration activity, which, in turn, induces apoptosis in breast cancer cells [33]. Another study stated that proteins isolated from lupin seeds arrested the G_0_/G_1_ phase of the cell cycle and induced apoptosis of cancer cells [55]. Thus, the presence of phenolic acids, flavonoids, flavonols, lupanine, and related compounds and high-quality proteins is the key factor to exaggerate colorectal cancer cells’ apoptosis by unprocessed flour extracts of lupin cultivars.

To explore the mechanistic insight of the significant colorectal cancer apoptotic activity of lupin seed flours, different cell-based bioassays were adopted, which revealed significant roles of lupin seed flours in the up-regulation of mitochondrial ROS generation as well as caspase-3/7 activation (Figure 7). Alike healthy cells, redox homeostasis is ubiquitously present in cancer cells to ensure maintenance of the persistent balance between the generation and neutralization of ROS. Extensive research on cancer biology reveals that malignancy is induced by severe oxidative-stressed conditions because of the triggering of genetic alteration, metabolic dysfunction, and mutagenic events by the constant formation of excessive ROS in cancer cells. Interestingly, malignant cells can alter metabolism to accelerate cell damage and apoptosis when excessive ROS is generated. These anticipated responses of cancer cells to excessive ROS facilitate the target elevation of ROS generation as a therapeutic option to induce apoptosis of cancer cells [27,56,57,58]. As lupin cultivars (especially Mandelup, Jurien, and Barlock) elevate the formation of mitochondrial ROS, apoptosis of colorectal cancer cells has been observed. A rich content of polyphenols, flavonoids, and flavonols has been identified in cultivars of lupin in previous research [2]. It is evidenced that phenolic compounds induce oxidative stress in cancer cells by elevating ROS generation through the Fenton reaction. Phenolics, at higher concentrations, initiate the chain redox reaction in the presence of oxygen, leading to the formation of oxygen radicals, which, in turn, react with metallic pro-oxidants to form hydrogen peroxide. The hydrogen peroxide undergoes a Fenton reaction to generate hydroxyl radicals, which prompt oxidative damage of cellular macromolecules, leading to apoptosis of cancer cells [40,59].

Caspases are basically aspartate-specific cysteinyl proteases that predominantly regulate cancer cell apoptosis. Caspases are considered a central cofactor in the initiation as well as the execution of apoptosis. The activation of caspases-2/8/9/10 initiates the apoptotic pathway, which is executed by caspases-3/6/7 through selective cleavage of DNA and other organelles of the cells, resulting in cell death. The success of the extrinsic and the intrinsic apoptosis exclusively mediated by the executor caspases causes extensive as well as irretrievable proteolytic activity of the caspases family [60,61]. Lupin cultivars Mandelup and Jurien showed significant caspases-3/7 activation in HCT116 and HT29 cell lines, respectively. Moreover, moderate activation of caspases-3/7 has also been observed by Barlock and Mandelup in HCT116 and HT29 cell lines, respectively. Hence, caspases-3/6/7 activation is the key factor to merge the initiation and completion of the apoptotic pathway, and colorectal cancer cells’ apoptosis is attained due to the activation of caspases-3/7 by lupin seed flours.

Taken together, it can be concluded from this investigation that lupin cultivars Mandelup, Barlock, and Jurien exhibit potential anticancer activity against HCT116 and HT29 colorectal cancer cell lines. Domestic heating leads to tremendous loss of anticancer activity of lupin cultivars. The mechanism behind this activity involves the activation of caspases-3/7 enzymes and the elevation of intracellular ROS in mitochondria, leading to triggering of the apoptosis of cancer cells. Thus, extensive research focusing on the unprocessed seeds of lupin cultivars, especially Mandelup and Jurien, might be helpful for discovering potent drugs with colorectal-cancer-attenuating potential.

## 5. Conclusions

In this investigation, seed flours obtained from Australian lupin cultivars (Mandelup, Barlock, and Jurien) showed significant anticancer activity on HCT116 and HT29 colorectal carcinoma cell lines. From the nine cultivars, 60% of cell proliferation was attained by Barlock, Jenabillup, Jurien, and Mandelup cultivars on the HCT116 cell line and by Jurien and Mandelup cultivars on the HT29 cell line. Significant loss of activity was observed following heat processing, which demonstrated that domestic cooking of lupin seeds deteriorates their anticancer activities. Furthermore, Annexin V-FITC assay indicated early and late apoptosis as major causes of decreasing colorectal carcinoma cell viability. Furthermore, Mandelup and Jurien showed significant elevation of mitochondrial ROS along with cleaving procaspases to active forms caspases-3/7, which are the central controllers of the initiation and completion of apoptosis of colorectal cancer cells.

Lupin cultivars are rich in phenolic compounds (phenolic acids, flavonoids, flavonols, and flavones), proteins, micro-metals, dietary fibers, alkaloids, fatty acid esters, and others. These bioactive compounds have the potential to initiate and successfully complete apoptosis through the increased generation of intracellular ROS and the modulation of caspases activities. In addition, multiple mechanisms may result in the synergism and up-regulation of colorectal cancer cell apoptosis. Therefore, it can be established from this study that lupin cultivars Mandelup and Jurien are capable of alleviating colorectal carcinoma cells by triggering the induction of apoptosis.

## Figures and Tables

**Figure 1 cells-12-02557-f001:**
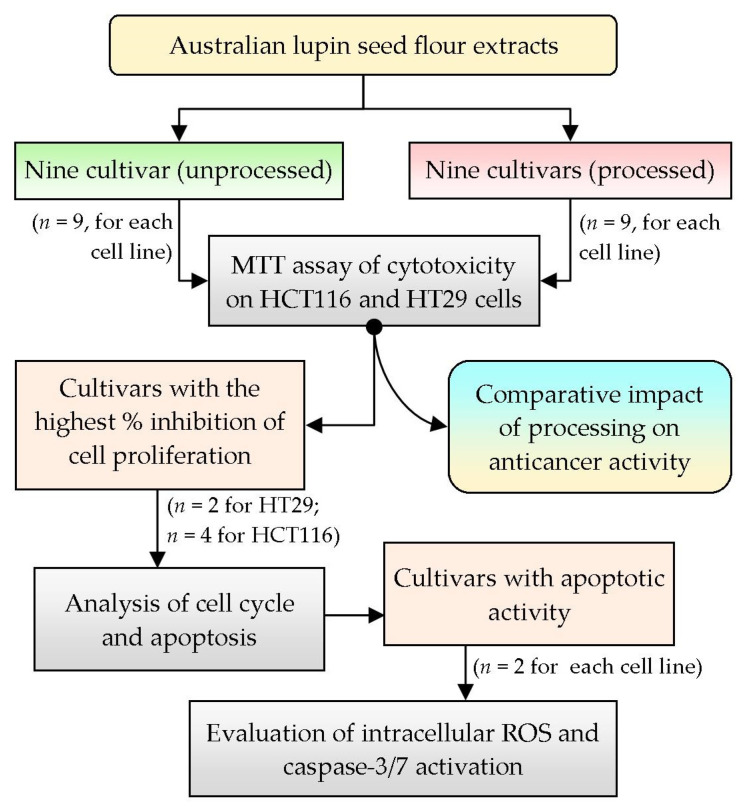
Overview of the summary of the research design and selection of methodology; *n*: number of cultivars tested.

**Figure 2 cells-12-02557-f002:**
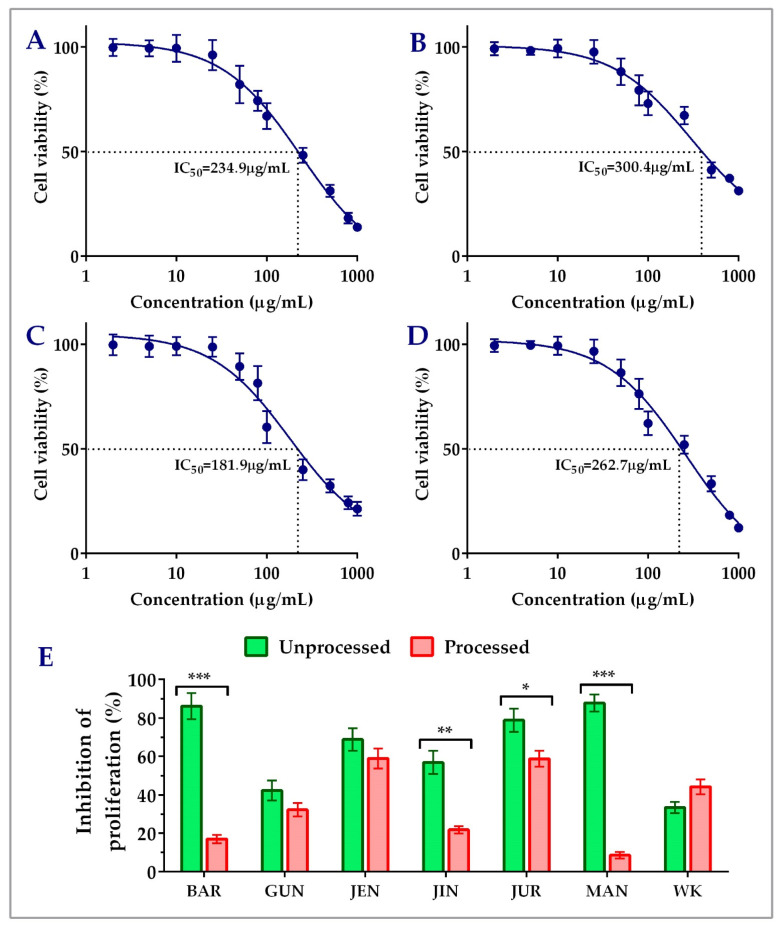
Cell viability of HCT116 colon carcinoma cells after treatment with lupin flour extracts (processed and unprocessed) for 24 h; cytotoxicity of unprocessed flour extracts of Barlock (**A**), Jenabillup (**B**), Jurien (**C**), and Mandelup (**D**) at various concentrations (0–1000 μg/mL) with IC50 values. (**E**) Comparison of inhibition of cell proliferation by unprocessed and processed flour extracts (1000 μg/mL). BAR: Barlock, GUN: Gunyidi, JEN: Jenabillup, JIN: Jindulee, JUR: Jurien, MAN: Mandelup, and WK: WK-338. Data are expressed as mean ± SEM, where *n* = 3. * *p* < 0.05 ** *p* < 0.01, and *** *p* < 0.001 indicate different levels of statistical significance, where processed and unprocessed groups of each cultivar were compared using Student’s *t*-test.

**Figure 3 cells-12-02557-f003:**
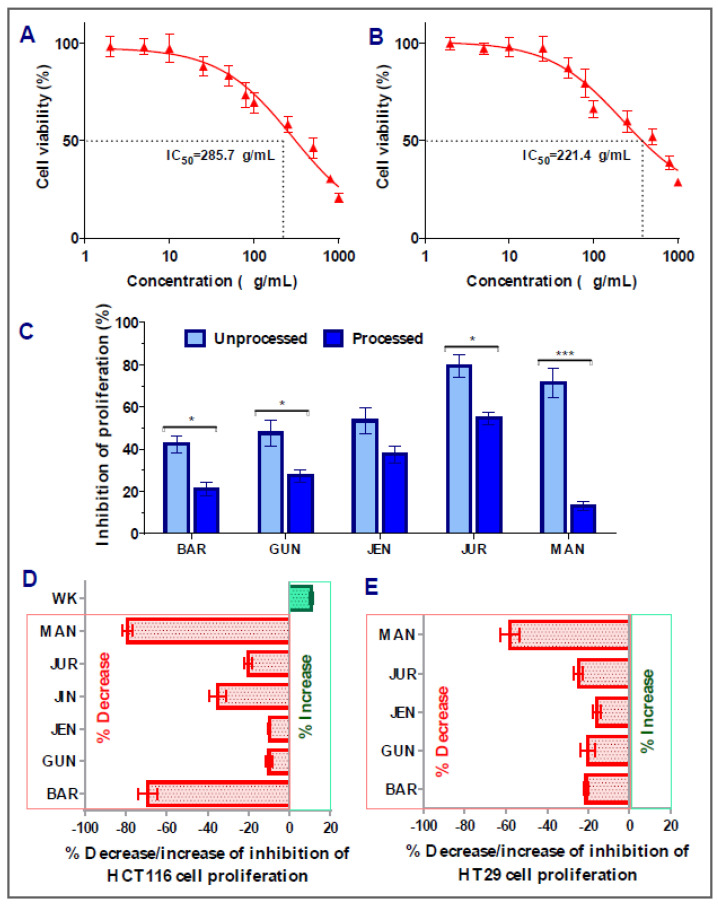
Cell viability of HT29 colon carcinoma cells after treatment with lupin flour extracts (processed and unprocessed) for 24 h; cytotoxicity of unprocessed flour extracts of Jurien (**A**), and Mandelup (**B**) at various concentrations (0–1000 μg/mL) with IC_50_ values. (**C**) Comparison of inhibition of cell proliferation by unprocessed and processed flour extracts (1000 μg/mL). (**D**,**E**) Impact of processing of lupin flours on colon cancer cytotoxicity. BAR: Barlock, GUN: Gunyidi, JEN: Jenabillup, JUR: Jurien, and MAN: Mandelup. Data are expressed as mean ± SEM, where *n* = 3. * *p* < 0.05 and *** *p* < 0.001 indicate different levels of statistical significance, where processed and unprocessed groups of each cultivar were compared using Student’s *t*-test.

**Figure 4 cells-12-02557-f004:**
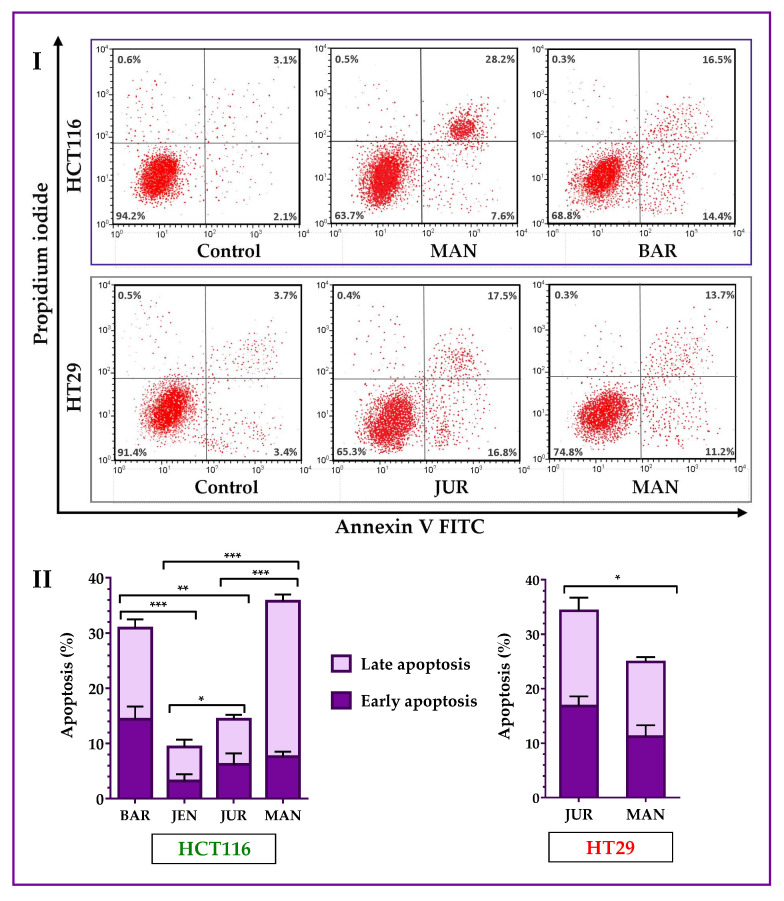
Apoptosis of HCT116 and HT29 colon carcinoma cells after treatment with lupin flour extracts (dose: 300 μg/mL) for 24 h. I indicates flow cytometry profiles of Annexin-V-FITC staining, and II indicates % apoptosis obtained from the Annexin V-FITC Apoptosis assay. BAR: Barlock, JEN: Jenabillup, JUR: Jurien, and MAN: Mandelup. Data are expressed as mean ± SEM, where *n* = 3. * *p* < 0.05, ** *p* < 0.01, and *** *p* < 0.001 indicate different levels of statistical significance, where comparisons among cultivars were analyzed using Student’s *t*-test.

**Figure 5 cells-12-02557-f005:**
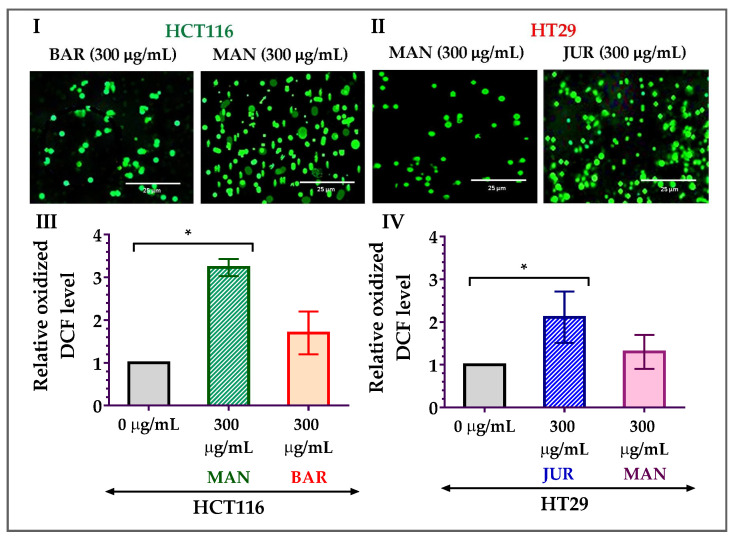
Generation of ROS in HCT116 and HT29 colon carcinoma cells after treatment with lupin seed flour extracts for 24 h. (**I**,**II**) Fluorescent images of HCT116 and HT29 cells, respectively. (**III**) Oxidized DCF level in HCT116 cells. (**IV**) Oxidized DCF level in HT29 cells. BAR: Barlock, JUR: Jurien, and MAN: Mandelup. Data are expressed as mean ± SEM, where *n* = 3. * *p* < 0.05 indicates statistical significance, where the ROS-generating effect of treated cells (300 μg/mL) was compared to that of untreated cells (0 μg/mL) using Student’s *t*-test.

**Figure 6 cells-12-02557-f006:**
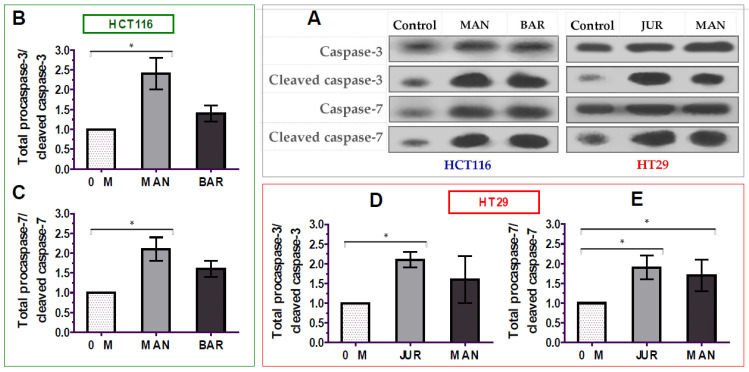
Activation of caspase-3/7 enzymes in HCT116 and HT29 colon carcinoma cells after treating with lupin flour extracts (300 μg/mL) for 24 h. (**A**) Immune blots representing three different experimental groups. (**B**,**C**) Caspase-3 and caspase-7 cleavage activity in HCT116 cells by different three experimental groups. (**D**,**E**) Caspase-3 and caspase-7 cleavage activity in HT29 cells by different three experimental groups. BAR: Barlock, JUR: Jurien, and MAN: Mandelup. Data were expressed as mean ± SEM, where *n* = 3. * *p* < 0.05 indicates statistical significance, where the caspase-3/7 cleaving effect of treated cells (dose: 300 μg/mL) was compared with that of untreated cells (0 μg/mL) using Student’s *t*-test.

**Figure 7 cells-12-02557-f007:**
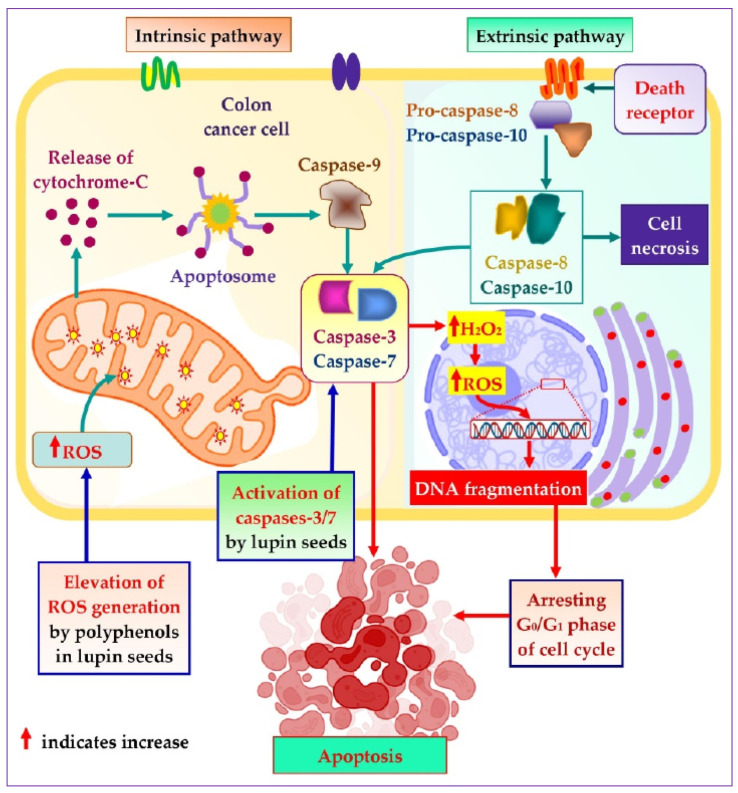
Mechanistic insight into apoptotic activity of lupin seeds on colorectal cancer cells by caspases-3/7 activation and mitochondrial ROS elevation.

## Data Availability

Not applicable.

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
