# Peer review of "Attenuating Colorectal Cancer Using Nine Cultivars of Australian Lupin Seeds: Apoptosis Induction Triggered by Mitochondrial Reactive Oxygen Species Generation and Caspases-3/7 Activation"

_cells, 2023, doi:10.3390/cells12212557_

Round 1

Reviewer 1 Report

Comments and Suggestions for Authors

Author Response

Dear reviewer, thank you very much for your valuable comments and suggestions.

We have studied nine cultivars of lupin seed fours on primary and advanced stage colorectal cancer cell lines. Another studies reported anticancer activities against another cancers. So far of our best knowledge, these nine cultivars were not studied previously for anticancer activities against colorectal cancer cell lines.

In future, we might include in vivo animal model study as well as if possible clinical trials in our next research project.

Minor points:

  1. In this work, the authors carry out an analysis of the cytotoxicity potential of various lupins extracts with consideration of the impact of heat processing on colon cancer cell lines (HCT116 and HT29). The study would have been more interesting if the authors had also compared the activity of lupin extracts on a healthy cell line.

Response: in this research we have focused on screening of the anticancer activities of lupin seed flours using colorectal cancer cell lines (both primary and advanced stage cancer cells). The non-toxic behavior of lupin seeds on healthy cell lines has already been reported by many of the researchers in different research contexts [1,2]. Thus, the healthy cell line study has been excluded to minimize the cost.

Reference: 

  1. Stobiecki, M.; Blaszczyk, B.; Kowalczyk‐Bronisz, S.; Gulewicz, K. The toxicity of seed extracts and their fractions from Lupinus angustifolius L. and Lupinus albus L. Journal of applied toxicology 1993, 13, 347-352.
  2. Escudero-Feliu, J.; García-Costela, M.; Moreno-SanJuan, S.; Puentes-Pardo, J.D.; Arrabal, S.R.; González-Novoa, P.; Núñez, M.I.; Carazo, Á.; Jimenez-Lopez, J.C.; León, J. Narrow Leafed Lupin (Lupinus angustifolius L.) β-Conglutin Seed Proteins as a New Natural Cytotoxic Agents against Breast Cancer Cells. Nutrients 2023, 15, 523.
  1. The results of the cell viability and apoptosis study were not presented well enough. I understand that the authors selected the most important data, but the results from the effect of other lupin extracts should also be presented, for example, in the supplementary materials. Please add these data.

Response: Considering all of the nine cultivars of Australian lupin seeds along with processed and unprocessed conditions, the number of samples were huge (n=18 for single testing). Besides, the cell anti-proliferative activity of all the cultivars was not significantly high. Anti-proliferative activity of processed flours was mild. Thus, based on the outcomes of preliminary study, only the cultivars with significant activities were selected for further studies including apoptosis analysis.

  1. Line – 216. Please add on the graphs in the legend the names of the lupin extracts. Figure 2 (BAR, JEN, JUR, MAN) and Figure 3 (JUR, MAN).

Response: The full form of the name of lupin cultivars in the legend of Figure 2 and Figure 3 were added in captions of the figures (please see line number 240-243, and 262-266).

  1. Figure 2 E. Why did the authors choose the highest concentration used in the study (1000μM) for comparison of inhibition of cell proliferation by unprocessed and processed flour extracts? After all, a concentration of 300 μM was chosen for the next stages of the study.

Response: As we tested crude extracts, the cell proliferation study was conducted by utilizing dose dependent manner. Various concentrations ranging from 0 μg/mL to 1000 μg/mL were selected. We found the IC50 values of the extracts were ranging from 180 μg/mL to300 μg/mL. Therefore, a concentration of 300 μg/mL was chosen for the next studies of the research.

  1. Figure 2 E. How do the authors explain the inverse dependence of proliferation inhibition in the case of unprocessed and processed WK extract?

Response: As stated in the discussion section (line number 446-449), about 10% increase of cytotoxic activity of WK-338 might be due to numerous possible reasons, such as (i) presence of heat stable bioactive metabolites with anticancer activities, (ii) presence of multiple bioactive compounds capable of undergoing heat-mediated chemical reactions resulting synergism, (iii) heat triggered chemical reactions among metabolites resulting in products with enhanced anticancer activity, and there could be other possibilities.

  1. In the discussion, the authors focus on the chemical composition of lupin and describe the most important active compounds determined by the GC-MS method in previous studies. It would be very interesting to add in this work a table comparing the most important lupin compounds before and after processing, performed by quantitative HPLC method in order to evaluate the loss of these compounds after the boiling process.

Response: We have already analyzed phytochemical profiling of processed and unprocessed flours of nine cultivars of Australian lupin by GC-MS [3]. However, we are interested to extend the metabolites screening of lupin cultivars by LC-MS analysis in near future.

Reference: 3. Mazumder, K.; Nabila, A.; Aktar, A.; Farahnaky, A. Bioactive Variability and In Vitro and In Vivo Antioxidant Activity of Unprocessed and Processed Flour of Nine Cultivars of Australian lupin Species: A Comprehensive Substantiation. Antioxidants 2020, 9, 282.

Reviewer 2 Report

Comments and Suggestions for Authors

Authors well described that the extraction of Lupin Seed Flour may induce the apoptosis in colon cancer cells. However, it is very important that which component can show different effects on cell viability. Only the effects of extraction on the cell viability is depleted in scientific sounds. It can be thought that 100 µM is so high concentration as to induce cell toxicity.

Author Response

Dear reviewer, we would like to appreciate your kind comments and suggestions. 

Please find our response as below:

Response: We screened the anticancer activities of the seed extracts of nine cultivars of Australian lupin and explored mechanistic insights too. For the first approach, in this project we attempted to evaluate the anti-colon cancer potentials of lupin seed flour extracts from the nine cultivars. After ensuring available funding and facilities, in future we will also take initiative for  isolation and analysis of metabolites to identify lead compounds from lupin cultivars. 

As we tested crude extracts, the cell proliferation study was conducted by utilizing dose dependent manner with various concentrations ranging from 0 μg/mL to 1000 μg/mL. We found the IC50 values of the extracts were ranging between 180 and 300 μM.

Reviewer 3 Report

Comments and Suggestions for Authors

The study was focused on 9 cultivars of australian 2 lupin seeds. Human colon cancer cell lines were treated by lupin flour extracts with varying concentrations of 1.0-1000 μg/mL for 48 hours.  I have few comments. Authors concluded cell cycle arrest although no cell cycle analysis was performed. Moreover, the induction of apoptosis after treatment was debatable (see below).  

Line 105, 106, 128, etc.: correct the symbol (100◦C) (100 °C)

Line 108: The speed centrifugation should be in g not in rpm. Change (4000 rpm)

Line168: Please, correct the endpoint of Reactive Oxygen Species Detection. It is impossible after seeding cells on 96 well plate to analyse cells on flow cytometer. Moreover, the authors showed fluorescent images (line 277).

Line 178: The excitation and emission are not correct 485 and 530 nm. I assume that the author used 488 nm laser and detector 528nm.  

Line 210: 42-79%. 42~79%.

Line 212:  Correct the sentence. It is unclear. „About >60% of inhibition was obtained  by JUR, and MAN with IC50 values ranging from 221 ± 382 μM.“

Line 223, etc.: Student’s t-test. student’s t-test.

Fig3: The axis label is confused, change it. „% Change of inhibition of HCT116 cell proliferation”

Line 252: The cell analysis was not performed only detection of cell death by Annexin V.

Line 293, Fig6: The cleaved fragments of caspases are strong. Please, can you add the whole membranes (raw data) from 3 independent Western blots?

Author Response

Dear reviewer, we would like to appreciate for your kind conservation, comments and suggestions. we tried our best to follow you as much as possible. Please find our response as below:

Comments: The study was focused on 9 cultivars of australian 2 lupin seeds. Human colon cancer cell lines were treated by lupin flour extracts with varying concentrations of 1.0-1000 μg/mL for 48 hours.  I have few comments. Authors concluded cell cycle arrest although no cell cycle analysis was performed. Moreover, the induction of apoptosis after treatment was debatable. 

Line 105, 106, 128, etc.: correct the symbol (100◦C) (100 °C)

Response: Sorry for the mistaken. The symbol was corrected in the respected lines as per suggestion. Please check line number 110, 112, 116, 117, and 141

Line 108: The speed centrifugation should be in g not in rpm. Change (4000 rpm)

Response: The speed of centrifugation was converted and corrected to g instead of rpm. Please check line number 119 and 203.

Line168: Please, correct the endpoint of Reactive Oxygen Species Detection. It is impossible after seeding cells on 96 well plate to analyse cells on flow cytometer. Moreover, the authors showed fluorescent images (line 277).

Response: The cells were seeded on 24-well plate during the experiment. Mistakenly 96-well was mentioned in the manuscript. The error was corrected in line number 188 as well.

Line 178: The excitation and emission are not correct 485 and 530 nm. I assume that the author used 488 nm laser and detector 528nm.

Response: Sorry for the typographical error. We used 488 nm laser and detector 528 nm. It was corrected in line number 196 (please check).

Line 210: 42-79%. 42~79%.

Response: The sign was corrected in line number 249 as per recommendation.

Line 212:  Correct the sentence. It is unclear. „About >60% of inhibition was obtained  by JUR, and MAN with IC50 values ranging from 221 ± 382 μM.“

Response: The sentence was re-structured for clear understanding (please see line number 251-152).

Line 223, etc.: Student’s t-test. student’s t-test.

Response: The error was corrected in line number 246, 268, 274, 329, 354, and 382.

Fig3: The axis label is confused, change it. „% Change of inhibition of HCT116 cell proliferation”

Response: As per recommendation, the x-axis label was changed. Please see Figure 3.

Line 252: The cell analysis was not performed only detection of cell death by Annexin V.

Response: Sorry for the error. The methodology and results regarding the study were corrected.

Line 293, Fig6: The cleaved fragments of caspases are strong. Please, can you add the whole membranes (raw data) from 3 independent Western blots?

Response: Raw data of Western blots was added as Figure S1 in Supplementary file.

Reviewer 4 Report

Comments and Suggestions for Authors

The paper is interesting but needs improvement.

Line 127: "1%" refers to the concentration of the solution, not the actual final concentration of the antibiotics. The final concentrations of penicillin and streptomycin are, respectively 100U/mL and 100µg/mL.

Lupin seed extract is a mixture of many ingredients and its concentration can be expressed asµg/mL”, not “µM”. This should be corrected in the text, figures and abstract.

Author Response

Dear reviewer, we would like to appreciate your kind comments and suggestions. In response to your comments please find our response as below.

Comments: The paper is interesting but needs improvement.

Line 127: "1%" refers to the concentration of the solution, not the actual final concentration of the antibiotics. The final concentrations of penicillin and streptomycin are, respectively 100U/mL and 100µg/mL.

Response: The concentrations of penicillin and streptomycin were corrected as suggested (please check line number 139-140).

Lupin seed extract is a mixture of many ingredients and its concentration can be expressed as “µg/mL”, not “µM”. This should be corrected in the text, figures and abstract.

Response: The unit of concentration was corrected to µg/mL instead of µM in the text, figures and abstract.

Round 2

Reviewer 2 Report

Comments and Suggestions for Authors

I hope the further study elucidate which specific compound has the apoptosis inducing effects.